# Rankmax: An Adaptive Projection Alternative to the Softmax Function

**Weiwei Kong**[*]
Georgia Institute of Technology
wwkong@gatech.edu

**Walid Krichene**
Google Research
walidk@google.com

**Nicolas Mayoraz**
Google Research
nmayoraz@google.com

**Steffen Rendle**
Google Research
srendle@google.com

**Li Zhang**
Google Research
liqzhang@google.com

## Abstract

Many machine learning models involve mapping a score vector to a probability vector. Usually, this is done by projecting the score vector onto a probability simplex, and such projections are often characterized as Lipschitz continuous approximations of the argmax function, whose Lipschitz constant is controlled by a parameter that is similar to a softmax temperature. The aforementioned parameter has been observed to affect the quality of these models and is typically either treated as a constant or decayed over time. In this work, we propose a method that adapts this parameter to individual training examples. The resulting method exhibits desirable properties, such as sparsity of its support and numerically efficient implementation, and we find that it significantly outperforms competing non-adaptive projection methods. In our analysis, we also derive the general solution of (Bregman) projections onto the $(n, k)$–simplex, a result which may be of independent interest.

## 1 Introduction

The goal of many machine learning models, such as multi-class classification or retrieval, is to learn a conditional probability distribution. Such models often involve projecting a vector on the probability simplex, and a general form of such a projection is given by

$$p_\alpha(z) = \underset{x \in \Delta^{n-1}}{\operatorname{argmin}} \left\{ -\langle z, x \rangle + \frac{1}{\alpha} g(x) \right\} \tag{1}$$

where $z \in \mathbb{R}^n$ is the vector that is being projected, $\Delta^{n-1}$ is the standard probability simplex, and $g$ is a strongly convex function. For example, in the neural network setting, $z$ can be the output of the last hidden layer, an unconstrained vector, and $p_\alpha(z)$ can be the final output of the network, a probability distribution. Problem (1) has been studied extensively. It reduces to the Euclidean projection [27, 12] when $g$ is the squared Euclidean norm, and the entropy projection [10, 5, 3] when $g$ is the negative entropy. In the latter case, $p_\alpha$ is the widely used softmax function.

The function $p_\alpha$ can be viewed as a continuous approximation of the argmax function, and the parameter $\alpha$ directly controls its Lipschitz constant. It corresponds, for example, to the inverse temperature in the softmax (or Gibbs) distribution. It is often treated as a constant [33, 30] or a calibration parameter [13], but it can also be varied over time, according to a fixed schedule. One of

---

[*]Work done during an internship at Google Research.

the earliest examples is in simulated annealing for discrete optimization [19, 28, 14, 6], where the temperature of a Gibbs distribution is decreased over time. This has also been studied in the context of smooth stochastic optimization [22, 9], and recent work [31, 17, 32] shows empirical evidence that a time-varying $\alpha$ can improve model quality. The existing approaches use a fixed schedule for $\alpha$, and a natural question is whether $\alpha$ can be made adaptive to individual training examples. To the best of our knowledge, this has not yet been attempted.

In our derivation of an adaptive projection method, we consider a more general setting than problem (1). Instead of projecting on the simplex, we project on the $(n, k)$ simplex, the intersection of the simplex with a hypercube. This is motivated by applications to multi-class classification and retrieval, where the goal is to select $k$ items out of $n$. The $(n, k)$–simplex has a richer combinatorial structure than the standard simplex, and has been studied in online learning [36, 18, 20, 2] and convex optimization [29, 34].

Our first contribution is to derive a general solution of the projection when $g$ is separable, a property that holds for the commonly-used projections. More precisely, we show that the corresponding KKT conditions reduce to a parameter search over a monotone one–dimensional problem (regardless of the value of $n$ or $k$) that can be solved using a simple bisection method. This unifies and generalizes several special cases, including the projection on the standard simplex ($k = 1$) derived in [12, 21, 7], and several other special cases in the entropy and the Euclidean case [36, 37, 1].

Our second and main contribution is to propose a method with adaptive $\alpha$ for classification problems. Our method is derived in the Euclidean case, which is particularly attractive as the solution reduces to a thresholding operation, and we show a one-to-one correspondence between $\alpha$ and the threshold. This motivates a particular form of adaptive $\alpha$, which results in several desirable properties: the projection can be computed in closed-form, and its support becomes sparser on examples that have a lower classification error. Finally, we test our method on a popular recommendation problem to give some insights into its empirical performance. The results indicate that the adaptivity can give significant performance gains.

**Organization of the paper**  We first derive a general solution of the projection in Section 2. Then, we consider a classification setting in Section 3 and present the adaptive projection and its properties. In Section 4, we present numerical experiments, and close with concluding remarks in Section 5. The proofs of our results are deferred to the supplement.

**Notation**  The set of real numbers is denoted by $\mathbb{R}$. The set of strictly positive real numbers is denoted by $\mathbb{R}_+$. The nonnegative part of a scalar $\alpha \in \mathbb{R}$ is denoted by $(\alpha)_+ = \max\{0, \alpha\}$. The inner product of two vectors $x, y \in \mathbb{R}^n$ is denoted by $\langle x, y \rangle$. The support of a vector $x \in \mathbb{R}^n$ is denoted by $\operatorname{supp}(x) = \{i \in \{1, \dots, n\} : x_i \neq 0\}$. We now consider a proper, lower semicontinuous, and convex function $f : \mathbb{R}^n \mapsto (-\infty, \infty]$. Let $\partial f$ denote the subdifferential of $f$, given by

$$\partial f(x) = \{v \in \mathbb{R}^n : f(u) \geq f(x) + \langle v, u - x \rangle, \forall u \in \mathbb{R}^n\} \quad \forall x \in \mathbb{R}^n.$$

Let $\operatorname{dom} f = \{x \in \mathbb{R}^n : f(x) < \infty\}$ denote the effective domain of $f$.

## 2  Projection on the $(n, k)$–simplex

We consider the following projection problem:

$$p_\alpha(z) = \underset{x \in \Delta_k^{n-1}}{\operatorname{argmin}} \left\{ -\langle z, x \rangle + \frac{1}{\alpha} g(x) \right\}, \tag{2}$$

where $z \in \mathbb{R}^n$ is a given vector, $g$ is a 1-strongly convex function, $\alpha > 0$ is a parameter, and the feasible set is the $(n, k)$–simplex given by $\Delta_k^{n-1} := \{x \in \mathbb{R}^n : \sum_{i=1}^n x_i = k, \ 0 \leq x \leq 1\}$. The $(n, k)$–simplex can be viewed as the convex hull of vertices of the $n$-hypercube that have exactly $k$ entries equal to 1. It arises for example in combinatorial online learning [36, 18, 20, 2]. It is also sometimes referred to as the capped simplex [36], since it is the intersection of the standard simplex scaled by $k$, with the $n$-hypercube. Our motivation in studying this problem comes form applications to classification and information retrieval, as discussed in the next section. Figure 1 gives an illustration of the $(3, 2)$–simplex. We also note that problem (2) is equivalent to the Bregman projection with distance generating function $g$, in a sense that is made more precise in the supplement.

## 2.1 A Lipschitz continuous approximation of $k$-argmax

It can be shown that for certain functions $g$, the function $p_\alpha(z)$ is a Lipschitz continuous approximation of the (generally discontinuous) $k$-argmax function given below.

$$p_\infty(z) := \operatorname*{argmax}_{x \in \Delta_k^{n-1}} \langle z, x \rangle. \tag{3}$$

The following proposition makes the previous statement more precise.

**Proposition 1.** *Suppose $g$ is 1–strongly convex and $\Delta_k^{n-1} \subseteq \operatorname{dom} g$. Then the following properties hold for any $z \in \mathbb{R}^n$ and $\alpha > 0$:*

(a) *$\lim_{\alpha \to \infty} p_\alpha(z) \in p_\infty(z)$;*

(b) *for any $1 \leq i, j \leq n$, we have that $p_\alpha(z)_i \geq p_\alpha(z)_j$ if and only if $z_i \geq z_j$;*

(c) *the function $p_\alpha$ is $\alpha$–Lipschitz continuous on $\mathbb{R}^n$.*

Intuitively, $p_\infty(z)$ is a function that assigns probability 1 to the top $k$ coordinates $i$ that maximize $z_i$, and part (a) states that this is the case in the limit $\alpha \to \infty$, regardless of $g$. Note that $p_\infty$ is set-valued since the maximizer is in general not unique, i.e. when there are ties in the entries of $z$. In contrast, for any finite $\alpha$, the maximizer of (2) is unique by strong convexity of $g$. Part (b) states that the projection is component-wise monotone, in the sense that coordinates that have a higher score are assigned a higher weight. Part (c) states that the parameter $\alpha$ directly controls the Lipschitz constant of the projection, a fact which motivates our adaptive method derived in Section 3.

## 2.2 Solution of the projection for separable $g$

We now give a characterization of the solution of (2) and a method to compute it, under the additional assumption that $g$ is separable. A characterization of the solution for a general function $g$ can be found in the supplement.

We first recall the definition of a separable function: a function $g : \mathbb{R}^n \to \mathbb{R}$ is said to be separable if there exists another function $h : \mathbb{R} \to \mathbb{R}$ such that for all $z$, we have $g(z) = \sum_{i=1}^n h(z_i)$. This condition holds for commonly used functions such as the squared Euclidean norm, the negative entropy, and the potential functions defined in [2]. We also recall the definition of the proximal operator: given a convex function $h$, the proximal operator of $h$ is given by

$$\operatorname{prox}_h(z) := \operatorname*{argmin}_{x \in \mathbb{R}^n} \left\{ h(z) + \frac{1}{2} \|u - z\|^2 \right\}. \tag{4}$$

The result below gives the characterization.

**Theorem 2.** *Suppose $g$ is 1–strongly convex and separable. and let $x, z \in \mathbb{R}^n$. Then, there exists $\mu \in \mathbb{R}$ such that*

$$x = p_\alpha(z) \quad \text{if and only if} \quad \begin{cases} \forall i, \ x_i = \Pi_{[0,1]}(y_i(\mu)), \\ \sum_{i=1}^n x_i = k, \end{cases} \tag{5}$$

*where $\Pi_{[0,1]}$ is the projection onto the interval $[0, 1]$ and the vector $y_i(\mu)$ has the following simple characterizations:*

(a) *(general case) we have*

$$y_i(\mu) = \begin{cases} \alpha(z_i - \mu) - v_0, & \text{if } \operatorname{prox}_h(\alpha(z_i - \mu)) \leq 0, \\ \operatorname{prox}_h(\alpha(z_i - \mu)), & \text{if } \operatorname{prox}_h(\alpha(z_i - \mu)) \in (0, 1), \\ \alpha(z_i - \mu) - v_1, & \text{if } \operatorname{prox}_h(\alpha(z_i - \mu)) \geq 1, \end{cases} \tag{6}$$

*for some $v_0 \in \partial h(0)$, $v_1 \in \partial h(1)$, and every $\mu \in \mathbb{R}$;*

(b) *(differentiable case) if $h$ is differentiable on its effective domain and $h' : \mathbb{R} \to \mathbb{R}$ is surjective, then we have*

$$y_i(\mu) = (h')^{-1}(\alpha(z_i - \mu)) \quad \forall \mu \in \mathbb{R}. \tag{7}$$

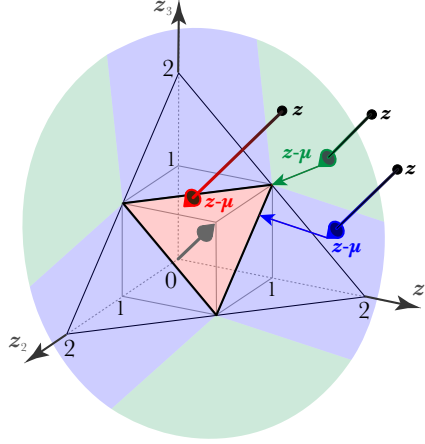

Figure 1: Illustration of the Euclidean projection on the $(3, 2)$–simplex, highlighted in red. Three example projections of vectors $z$ are given. As in eq. (8), the projection amounts to translating $z$ in the direction of the vector of all ones, followed by a projection on the unit hypercube.

Theorem 2 states that in order to solve the projection, it suffices to find a dual variable $\mu \in \mathbb{R}$ that satisfies the condition $\sum_{i=1}^{n} \Pi_{[0,1]}[y_i(\mu)] = k$, where $y_i(\mu)$ is given by (6) or (7), depending on differentiability of $h$. Observe, in particular, that $y_i(\mu)$ is a monotone function of $\mu$ in both cases. Indeed, under assumption (b), $h'$ is strictly increasing by strong convexity of $g$, and so is its inverse; under assumption (a), a similar, though more involved argument, is given in the supplement. Therefore, the function $\mu \mapsto \sum_i \Pi_{[0,1]}(y_i(\mu))$ is a monotone function of $\mu$ and an approximate solution can be computed using a bisection method. In other words, the projection is reduced to a one dimensional bisection problem regardless of the values of $n$ and $k$. Special cases of this result were derived in [37, 1] for the Euclidean and entropy projection, respectively. We also note that [25] derives an algorithm to compute the solution to a more general problem: projections on the permutahedron. Their algorithm MergeAndPool can be applied to compute projections on the $(n, k)$-simplex. However, Theorem 2 is more direct to obtain, and easier to interpret and implement.

We now specialize Theorem 2 to some common cases.

**Standard simplex**   When $k = 1$, case (b) is equivalent to the Bregman projection solution derived in [21, 7] (up to the change of variable described in Appendix A).

**Entropy projection**   Let $h(x) = x \log x$, defined for $x \geq 0$. Its derivative $h'(x) = 1 + \log x$ is surjective on $\mathbb{R}$ thus assumption (b) holds, and the projection is given by $x_i = \Pi_{[0,1]}(e^{\alpha(z_i - \mu) - 1})$, where $\alpha$ is the inverse temperature. With the change of variable $Z = e^{\alpha\mu + 1}$, this is equivalent to $x_i = \Pi_{[0,1]}(e^{\alpha z_i}/Z)$. For $k = 1$, this reduces to the softmax function. For $k > 1$, one can use the iterative algorithm in [36] to compute the normalization constant $Z$ in $\mathcal{O}(n^2)$ operations.

**Euclidean projection**   Let $g(x) = \frac{1}{2}\|x\|_2^2$. Then $h(x) = x$, assumption (b) applies, and the solution is

$$p_\alpha(z)_i = \Pi_{[0,1]}(\alpha(z_i - \mu)), \tag{8}$$

which has a simple geometric interpretation illustrated in Figure 1: the vector $\alpha z$ is first translated orthogonally to the hyperplane containing the simplex, then projected orthogonally onto the hypercube. This simple form of the Euclidean projection motivates the adaptive Rankmax method derived in the next section.

## 3   Adaptive projection for classification

In this section, we consider a multi-label classification setting. We denote the feature set by $\mathcal{X} \subset \mathbb{R}^d$ and the label set by $\mathcal{Y} = \{1, \ldots, n\}$, where $n$ is the number of classes. We are given a training set $T$

consisting of examples $(x, Y_x)$, such that for all $x$, the set $Y_x$ is a non-empty subset of $\mathcal{Y}$, potentially consisting of more than one label. For example, in information retrieval, $x$ represents a query and $Y_x$ is a set of documents relevant to that query. In item recommendation with implicit feedback [16], $x$ represents a user, and $Y_x$ is the set of items that the user interacted with.

A common modeling approach is to define a model that outputs a score vector $z_\theta : \mathcal{X} \to \mathbb{R}^n$, parameterized by $\theta \in \Theta \subseteq \mathbb{R}^r$, such that a higher score $z_\theta(x)_y$ indicates a higher relevance of the label $y$. The quality of the model is measured using retrieval metrics, such as Precision@$k$, defined as follows. Given an example[2] $(x, Y_x)$, we define

$$\text{Precision@}k(x) := \frac{1}{k} \left| \left\{ y \in Y_x : z_\theta(x)_y \geq z_\theta(x)_{[k]} \right\} \right|, \tag{9}$$

where $z_{[i]}$ denotes the $i^{\text{th}}$ largest element of $z$ (ties are broken arbitrarily). The condition $z_\theta(x)_y \geq z_\theta(x)_{[k]}$ simply means that the label $y$ has one of the top $k$ scores. Precision@$k$, and other similar metrics such as Recall@$k$, focus on the top $k$ scoring items, and are intended to reflect that a user will rarely look beyond the top retrieved items.

Projection on $\Delta_k^{n-1}$ is particularly well suited to this problem, and has been used in [23, 24, 1] for optimizing top-k metrics. Indeed, since the precision (9) is entirely determined by the set of top $k$ labels (regardless of the order within the set), it is natural to describe the problem in terms of selecting $k$ labels out of $n$. The set of subsets of size $k$ corresponds exactly to the extreme points of $\Delta_k^{n-1}$, making projections on $\Delta_k^{n-1}$ suitable for this formulation.

Finally, while the precision metric (9) is neither continuous nor differentiable in $\theta$, several approaches have been proposed to circumvent this, by optimizing a different loss as a proxy, often defined on the simplex. For example, [11] uses an approximation based on an optimal transport formulation; [35] uses a piece-wise linear approximation; [26] uses a quadratic loss; and one of the most widely used methods is to optimize the cross-entropy loss, or more generally, Fenchel-Young losses [7, 8].

## 3.1 Cross-entropy optimization

The cross-entropy[3] loss on the projected vector $p_\alpha(z_\theta(x))_y$ given by

$$\min_{\theta \in \Theta} \sum_{(x, Y_x) \in T} \sum_{y \in Y_x} - \log p_\alpha(z_\theta(x))_y. \tag{10}$$

Using cross-entropy as a proxy for top-k metrics can be motivated by the following consistency result. Suppose $p_\alpha$ is surjective (this is, for example, the case for the Euclidean projection). Then, for any example $(x, Y_x)$, if $z \in \text{argmin}_{z \in \mathbb{R}^n} \sum_{y \in Y_x} - \log p_\alpha(z)_y$, we have $z \in \text{argmax}_{z \in \mathbb{R}^n} |\{y \in Y_x : z_\theta(x)_y \geq z_\theta(x)_{[k]}\}|$. Indeed, if $z \in \text{argmin}_{z \in \mathbb{R}^n} \sum_{y \in Y_x} - \log p_\alpha(z)_y$, then there exists $c \in (0, 1]$ such that $p_\alpha(z)_y = c$ for every $y \in Y_x$, and $p_\alpha(z)_y \leq c$ for $y \notin Y_x$. By monotonicity of the projection (see Proposition 1(b)), we must then have $z_{y_+} \geq z_y$ for all $y_+ \in Y_x$ and $y \notin Y_x$ and thus $z$ maximizes the precision.

In other words, if there exists a classifier that perfectly fits a given training example $(x, Y_x)$ in the sense of (10), then such a classifier also has maximal precision (9) for that $x$. Of course, this has little implication for examples which are not perfectly classified. Nevertheless, this gives an intuitive justification for optimizing the cross-entropy loss (10) as a proxy for the recall metric (9).

## 3.2 Rankmax for the $(n, k)$–simplex

In this section, we derive an adaptive Euclidean projection function that we call Rankmax. Its primary goal is to make the parameter $\alpha$ in $p_\alpha$ (i.e., the Lipschitz constant of the projection) adapt to individual training examples. Another goal is to solve the following practical issue with the entropy formulation (10): the cross-entropy is undefined whenever $p_\alpha(z_\theta(x))$ assigns zero probability to a label $y \in Y_x$.

Before presenting the Rankmax function, we state some technical properties about the Euclidean projection. Recall that it has the simple form $p_\alpha(z)_i = \Pi_{[0,1]}(\alpha(z_i - \mu))$ given by equation (8). Moreover, the support of the projection is determined by $\mu$, since $i \in \text{supp}(p_\alpha(z))$ if and only if $z_i > \mu$. In particular, given a positive label $y$, if we can choose $\alpha$ such that $\mu < z_y$, then we guarantee that $y$ is in the support. In the result below, we establish the existence of such an $\alpha$ and show that it can be computed efficiently.

**Proposition 3.** *Fix $\eta > 0$, $z \in \mathbb{R}^n$, and a positive label $y \in \{1, \ldots, n\}$. Moreover, let*

$$\mu_y := \min\{z_y, z_{[k]}\} - \eta. \tag{11}$$

*Then, there exists $\alpha_y > 0$ such that $p_{\alpha_y}(z) = \Pi_{[0,1]}(\alpha_y(z - \mu_y))$. Furthermore, if $s := |\{i : z_i > \mu_y\}|$, then there exists $0 \le t < \min(s, k)$ such that*

$$\alpha_y = \frac{k - t}{\sum_{i=t+1}^{s}(z_{[i]} - \mu_y)}. \tag{12}$$

*Finally, the index $t$ can be computed in $\mathcal{O}(n \log k)$, as detailed in Algorithm 1 in the supplement.*

This allows us to define the following Rankmax projection function:

**Definition 4.** *Given a score vector $z \in \mathbb{R}^n$ and label $y \in \{1, \ldots, n\}$, the Rankmax projection function is given by*

$$Rankmax(z, y) = \Pi_{[0,1]}(\alpha_y(z_\theta(x) - \mu_y)) \tag{13}$$

*where $\mu_y$ is given by (11) with $\eta$ set to $1$ and $\alpha_y$ is given by (12).*

We now make several remarks about Proposition 3 and the definition of the Rankmax function. First, the indices $s$, $t$ have a simple interpretation: $s$ is the size of the support, and $t$ is the number of labels that are assigned weight $1$. In other words,

$$\text{Rankmax}(z, y)_{[i]} = \begin{cases} 1 & \text{if } i \in [1, t] \\ \alpha_y(z_{[i]} - \mu_y) & \text{if } i \in [t+1, s] \\ 0 & \text{if } i \in [s+1, n], \end{cases}$$

and thus eq (12) is just a rephrasing of the condition $\sum_{i=1}^{n} p_\alpha(z)_i = k$. Second, the choice of $\alpha$ given by the proposition makes the support of the projection *adapt to the positive label $y$*. Indeed, by definition of $\mu_y$, the label $y$ is guaranteed to be in the support since $z_y \ge \mu_y + \eta$, and labels that have a score below the $\eta$ margin are assigned zero weight. In particular, when the positive label is ranked higher, the support of the projection becomes smaller. Third, we choose a unit margin ($\eta = 1$) in the Rankmax definition since the projection is unchanged when $z$ and $\eta$ are simultaneously scaled by a constant $\gamma$. This can be seen from equations (11)-(12), where if $\mu_y$ is scaled by $\gamma$ and $\alpha_y$ is scaled by $1/\gamma$ then the product $\alpha_y(z - \mu_y)$ remains unchanged. Fourth, note that whenever $z_y \le z_{[k]}$, i.e., the positive label is not in the top $k$, we have $\text{Rankmax}(z, y)_y = \alpha_y(z_y - (z_y - 1)) = \alpha_y$, giving a concrete interpretation of the adaptivity. The parameter $\alpha_y$, which controls the Lipschitz constant of the projection, scales directly with the probability of the positive label $y$. To make an analogy with the temperature in softmax, this would correspond to decreasing the temperature for examples which have a lower loss.

It's also important to emphasize that both $\mu_y$ and $\alpha_y$, in the expression of the Rankmax function, depend on $y$, i.e., the projection depends on the positive label, as well as $\theta$. This is not an issue during training, but at first glance may seem problematic for inference, since the label is not available. However, since the task is to rank the labels, it suffices to compute the score vector $z_\theta(x)$ without a projection, since $z$ entirely determines the ranking; any projection $p_\alpha$ preserves ranks by monotonicity (Proposition 1(b)), regardless of $\alpha$. Another interpretation of the previous remark is that the projection is not part of the model, but instead part of the training procedure.

Finally, we give the expression of the cross-entropy loss under the Rankmax projection. Applying (10) to the Rankmax function, we obtain

$$\min_{\theta \in \Theta} \sum_{(x, Y_x) \in T} \sum_{y \in Y_x} -\log \min(1, \alpha_y(z_\theta(x)_y - \mu_y)). \tag{14}$$

It is worth mentioning that the Rankmax function can be used in losses aside from cross-entropy. One example is the set of Fenchel-Young (FY) losses [7, 8] which are of the form $L(z, y) =$

$f_z(y) - f_z(p_\alpha(z))$ where $p_\alpha$ is the projection defined in (2), $z$ is a score vector, $y$ is a label, and $f_z$ is a regularization function. In particular, one could consider replacing the static function $p_\alpha$ above with the adaptive Rankmax.

## 3.3 Rankmax for the standard simplex

In this section, we specialize the previous results to the standard simplex. Since $k = 1$, this simplifies the expressions of $\mu_y$ and $\alpha_y$. Indeed, equation (11) simplifies to $\mu_y = z_y - \eta$ (since we always have $z_y \leq z_{[1]}$), and equation (12) simplifies to $\alpha_y = 1/\sum_{i=1}^{n}(z_i - z_y + \eta)_+$, since $t \in [0, \min(s, k))$ and must be equal to 0. The Rankmax projection is then

$$\text{Rankmax}(z, y) = \frac{(z - z_y + 1)_+}{\sum_{i=1}^{n}(z_i - z_y + 1)_+}. \tag{15}$$

**Comparison to Softmax cross-entropy**

To illustrate the difference with the softmax function, we compare their cross-entropy losses (cf. (14)) and their corresponding gradients. Denote the cross-entropy losses

$$\ell_{\text{Rankmax}}(z, y) := -\log \text{Rankmax}(z, y)_y = \log \sum_{i=1}^{n}(z_\theta(x)_i - z_\theta(x)_y + 1)_+,$$

$$\ell_{\text{Softmax}}(z, y) := -\log \text{Softmax}(z)_y = -z_y + \log \sum_{i=1}^{n}\exp(z_\theta(x)_i).$$

Then,

$$\frac{\partial \ell_{\text{Rankmax}}(z, y)}{\partial z_i} = \begin{cases} -s\text{Rankmax}(z, y)_y, & \text{if } i = y, \\ \text{Rankmax}(z, y)_y, & \text{if } i \neq y \text{ and } z_i \geq z_y - 1, \\ 0, & \text{otherwise.} \end{cases} \tag{16}$$

where $s$ is the size of the support, $s = |\{i \in [1, n] : z_i > z_y - 1\}|$ as defined in Proposition 3, and

$$\frac{\partial \ell_{\text{Softmax}}(z, y)}{\partial z_i} = \begin{cases} -1 + \text{Softmax}(z)_y, & \text{if } i = y, \\ \text{Softmax}(z)_i, & \text{if } i \neq y. \end{cases}$$

We highlight some similarities and differences. In both cases, we have $\sum_{i=1}^{n} \partial \ell(z, y)/\partial z_i = 0$. In fact this is the case for all projections of the form given in (2), since for all $\lambda \in \mathbb{R}$, we have $p_\alpha(z + \lambda \mathbf{e}) = p_\alpha(z)$, where $\mathbf{e}$ is the vector of ones, i.e., $p_\alpha$ is constant along the direction $\mathbf{e}$, and thus its gradient is orthogonal to $\mathbf{e}$. One difference is that in the case of Rankmax, the support of the gradient is sparser when the positive label is ranked higher, potentially leading to faster updates. It is also interesting to observe that the gradient is entirely determined by the probability of the positive label $y$ and the support of the projection.

**Comparison to Sparsemax**

We now discuss how the Rankmax function compares to the Sparsemax loss proposed in [26]. Both use a Euclidean projection onto the $(n, k)$–Simplex in (8) under different choices of $\alpha$ and $k$. In particular, Sparsemax chooses $\alpha = 1$ and $k = 1$, whereas Rankmax chooses $\alpha$ adaptively and applies to any $k$ in $[1, n]$. On the other hand, for any label $y \in \mathcal{Y}$, the Rankmax projection assigns, by design, a positive probability to $y$, whereas the Sparsemax function may assign zero probability. Finally, the evaluation of the Rankmax and Sparsemax functions require $\mathcal{O}(n \log k)$ and $\mathcal{O}(n \log n)$ operations, respectively (see [26, Algorithm 1] and the supplement).

**Connection with pairwise losses**

In the standard simplex case, the Rankmax projection is reminiscent of pairwise losses. For instance, the Weighted Approximate-Rank Pairwise (WARP) loss [35, 38] can be written, in our notation, as $\ell_{\text{WARP}}(z, y) = \sum_{i=1}^{n} \frac{w_s}{s}(z_i - z_y + 1)_+$ where $w$ is a fixed vector of increasing weights, and $s$ is the size of the support. Comparing this to equation (15), we see that $\ell_{\text{WARP}}(z, y) = w_s/(s\text{Rankmax}(z, y)_y)$, and its gradient is

$$\frac{\partial \ell_{\text{WARP}}(z, y)}{\partial z_i} = \begin{cases} -w_s & \text{if } i = y, \\ \frac{w_s}{s} & \text{if } i \neq y \text{ and } z_i > z_y - 1, \\ 0 & \text{otherwise.} \end{cases}$$

Comparing this to the expression of the Rankmax loss gradient in (16), we see that both have the same direction, but different magnitudes. For WARP, both the direction and magnitude of the gradient are entirely determined by $s$, which is the number of negatives above the margin. For Rankmax, the magnitude depends on the probability of $y$: for a fixed $s$, the magnitude of the gradient is proportional to $\text{Rankmax}(z, y)_y$, but since $\text{Rankmax}(z, y)_y = \alpha_y$, this magnitude is also proportional to the adaptive parameter $\alpha_y$.

|  | Movielens 100k | Movielens 20M | Movielens 1B |
|---|---|---|---|
| # examples (users) | 600 | 138K | 2.2M |
| # labels (movies) | 9K | 26K | 849K |
| # example-label pairs | 101K | 20M | 979M |

Table 1: Characteristics of datasets used.

## 4 Numerical experiments

In our experiments, we studied how well Rankmax performs as a multilabel classification loss, and compared it to both Softmax and Sparsemax [26]. For evaluation, we chose a recommender system task where the goal is to learn which movies (=labels) to recommend to a user (=example). We experimented with Movielens datasets [15], namely the datasets of 100K, 20M, and 1B ratings, the latter being artificially generated from the 20M dataset [4]. Basic statistics about the datasets are summarized in Table 1. The datasets were partitioned into 80% training, 10% cross-validation and 10% test. All models were trained using the first part of this split. The models were then used to rank all movies for each user. The resulting ranking is compared to the movies in the test set (=relevant movies) and we compute three metrics: (1) Recall@$K$ (R@$K$) measures how many of the relevant movies appear in the top $K$. When $K$ is large, this metric is particularly useful to assess the retrieval capabilities of a model. (2) Accuracy measures how often the very top ranked movie was actually relevant, and is useful for classification tasks. (3) AveragePrecision@$K$ (AP@$K$) assigns a decaying weight to each rank, so it is affected by the order within the top $K$ items, and favors models which rank relevant items higher. For all metrics, the higher the value, the better the quality. Hyper-parameters were tuned based on the cross-validation set. A detailed description of the experimental setting can be found in the supplement. Bolded numbers indicate the best performing loss function for a particular metric.

| Loss Function | ML 100K | | | ML 20M | | | ML 1B |
|---|---|---|---|---|---|---|---|
| | AP@10 | Accuracy | R@100 | AP@10 | Accuracy | R@100 | R@1000 |
| Rankmax | **0.154** | **0.342** | 0.387 | **0.210** | **0.397** | **0.492** | 0.0116 |
| Softmax | 0.147 | 0.300 | **0.401** | 0.182 | 0.342 | 0.483 | **0.0117** |
| Sparsemax | 0.147 | 0.286 | 0.384 | 0.191 | 0.369 | 0.464 | – |

Table 2: Qualitative comparison of loss functions

The results are reported in Table 2. We see that Rankmax improves over Softmax and Sparsemax on all metrics for the mid-size Movielens 20M dataset. On the smaller dataset, the metrics are noisier, and the difference between the different losses is less clear. It appears however that Rankmax has slightly better precision but worse recall. On the larger dataset we focused on large values of $K$ as the metrics for smaller $K$ were noisy. On R@1000, we did not find any noticeable difference between Rankmax and Softmax, and both algorithms take the same amount of time for each training step. Our implementation of Sparsemax is much slower than Rankmax and Softmax, mainly because it requires sorting the vector of scores $z_\theta(x)$. We did not succeed in running Sparsemax on the largest dataset ($n = 849$K).

We also carried out numerous experiments on Movielens 20M using a Softmax loss with temperature annealing according to an exponential decay, as in [17, 32]. Treating the final temperature as an additional hyperparameter, a joint hyperparameter search of the learning rate, the final temperature, and the regularization consistently indicated that temperature annealing does not improve the results on this particular dataset.

Figure 2 illustrates the evolution of AP@10 and R@100 over the course of training, for different learning rates. We plot these metrics as a function of the number of epochs to avoid any implementation specific effects. Wall-time plots for the same experiment are given in the appendix.

On the smaller dataset, the retrieval metrics improve rapidly using Rankmax but also deteriorate more significantly after some epochs, while this pattern is less pronounced for Softmax or Sparsemax (at least not within 20,000 epochs), suggesting that the use of Rankmax should be combined with an early stopping criterion. On Movielens 20M, Rankmax produces a $15\%$ improvement over Softmax and a $8\%$ improvement over Sparsemax across a range of learning rates. For this dataset, Rankmax is a better choice for any computational budget.

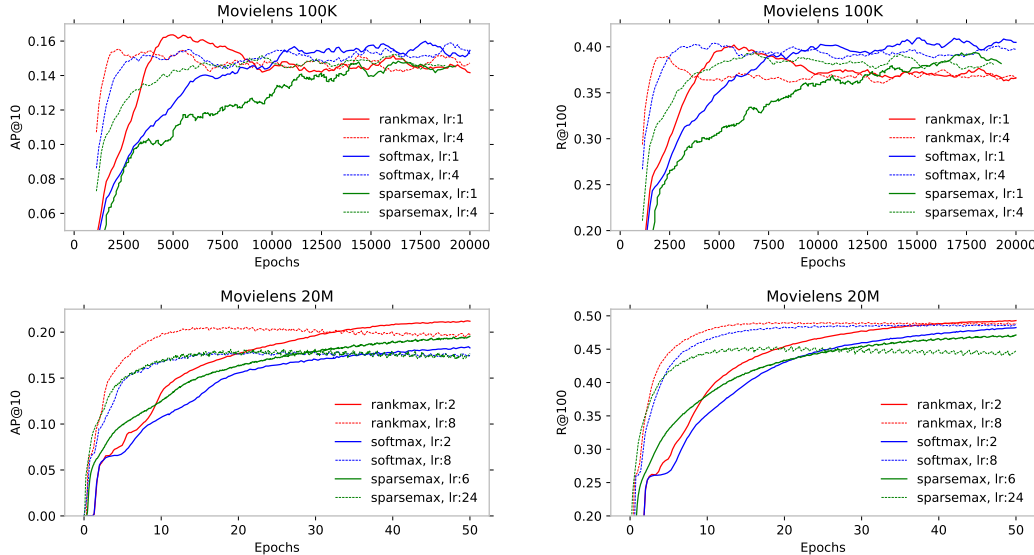

Figure 2: Evolution of retrieval metrics on MovieLens 100K and 20M over training epochs.

# 5 Concluding remarks

We derived an adaptive Euclidean projection method motivated by multi-label classification problems. The method adapts the parameter $\alpha$ to individual training examples, and shows good empirical performance.

Under cross-entropy loss, Rankmax is closely related to the pairwise losses as discussed in Section 3.3. While pairwise losses do not immediately fit into the projection framework of equation (1), this connection suggests that they may be closely related, and we believe this merits further investigation.

While we focused our discussion on the cross-entropy loss, the Rankmax projection can be used with other losses. Combining the adaptivity of Rankmax with the Fenchel-Young losses [7, 8] is an interesting direction for future work.

## Broader Impact

We derived a training method that is theoretically motivated and that shows a good performance on a popular benchmark. It can be used to expand the toolbox of practitioners, and may potentially lead to an improved model quality in some applications. Similarly to other optimization methods, the method we develop is not specific to a particular model or application.

## Footnotes

[2]Equation (9) defines the precision for one example. To measure the quality on the training or test set, it can be averaged over that set.

[3]The vector $p_\alpha$ should be scaled by $1/k$ to be a probability vector, but we omit this term as it only adds a constant to the loss.

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
