[Supplementary Material]

# Rankmax: An Adaptive Projection Alternative to the Softmax Function Supplementary Material

**Weiwei Kong**[*]
Georgia Institute of Technology
wwkong@gatech.edu

**Walid Krichene**
Google Research
walidk@google.com

**Nicolas Mayoraz**
Google Research
nmayoraz@google.com

**Steffen Rendle**
Google Research
srendle@google.com

**Li Zhang**
Google Research
liqzhang@google.com

## Abstract

This document consists of results that support the material in the paper "*Rankmax: An Adaptive Projection Alternative to the Softmax Function*", hereafter referred to as the main paper. It is assumed that the reader is already familiar with the notation and definitions in the main paper.

**Additional notation**. The next quantities are for a closed convex set $Z \subseteq \mathbb{R}^n$ and a proper, lower semicontinuous, and convex function $f$. Let $\delta_Z$ denote the convex indicator function of $Z$ where $\delta_Z(z) = 0$ if $z \in Z$ and $\delta_Z(z) = \infty$ if $z \notin Z$. Let $N_Z$ denote the normal cone of $Z$, given by

$$N_Z(z) := \{u \in \mathbb{R}^q : \langle u, \widetilde{z} - z \rangle \leq 0, \forall \widetilde{z} \in Z\} = \partial \delta_Z(z) \quad \forall z \in Z.$$

Let $\Pi_Z$ denote the projection onto the set $Z$ given by $\Pi_Z(z) = \operatorname{argmin}_{u \in Z} \|u - z\|^2/2$ for every $z \in \mathbb{R}^n$. Let $\operatorname{id}$ denote the identity operator given by $\operatorname{id}(z) = z$ for every $z \in \mathbb{R}^n$. Let $f^*$ denote the convex conjugate of $f$ given by $f^*(z) = \sup_{\widetilde{z} \in \mathbb{R}^n}\{\langle \widetilde{z}, z \rangle - f(\widetilde{z})\}$ for every $z \in \mathbb{R}^n$.

## A  On the equivalence with Bregman projections

Recall that the Bregman divergence [2] with a differentiable distance generating function $g$ is given by

$$D_g(x, y) = g(x) - g(y) - \langle \nabla g(y), x - y \rangle,$$

and the Bregman projection of a vector $\tilde{z}$ on $\Delta_k^{n-1}$ is given by

$$\operatorname*{argmin}_{x \in \Delta_k^{n-1}} D_g(x, \tilde{z}) = \operatorname*{argmin}_{x \in \Delta_k^{n-1}} \{g(x) - \langle \nabla g(\tilde{z}), x \rangle\}. \tag{17}$$

This is equivalent to (2) when $\nabla g(\tilde{z}) = \alpha z$. This identity is guaranteed to have a solution for all $z$ by strong convexity of $g$. Indeed by Theorem 23.5 in [7], $\alpha z \in \partial g(\tilde{z})$ if and only if $\tilde{z}$ maximizes $\langle \alpha z, \tilde{z} \rangle - g(z)$, but since $g$ is strongly convex, the latter always has a unique maximizer. Note that the maximization is over all of $\mathbb{R}^n$, unlike problem (2) where the minimization is over $\Delta_k^{n-1}$.

For example, when $g(x) = \frac{1}{2}\|x\|_2^2$, we have $D_g(x, y) = \frac{1}{2}\|x - y\|_2^2$ and $\nabla g(x) = x$. Thus, (17) and (2) are equivalent with $\tilde{z} = \alpha z$. When $g(x) = \sum_{i=1}^n x_i \log x_i$ with effective domain $\{x \in \mathbb{R}^n : x \geq 0\}$, the function $D_g$ is the (un-normalized) K-L divergence $D_g(x, y) = \sum_{i=1}^n x_i \log(x_i/y_i) + (y_i - x_i)$, defined on $\mathbb{R}_+^n \times \mathbb{R}_+^n$, and $\nabla g(x) = (1 + \log x_i)_{i=1 \ldots n}$ (note that $\nabla g$ is a bijection from $\mathbb{R}_+^n$ to $\mathbb{R}^n$). Thus (17) and (2) are equivalent up to the change of variable $\tilde{z}_i = e^{\alpha z_i - 1}$.

---

[*]Work done during an internship at Google Research.

# B  Properties of $p_\alpha$

This section presents several technical results regarding the function $p_\alpha$ from the main paper.

Before proceeding, we first present the following results on max functions, whose proofs can be found in [4].

**Lemma 1.** *Suppose that for some closed $X \times Y \subseteq \mathbb{R}^n \times \mathbb{R}^n$ and $\mu > 0$ we have a real–valued function $\Psi : \mathbb{R}^m \times \mathbb{R}^n \mapsto \mathbb{R}$ that satisfies:*

    *(A1)  $-\Psi(x, \cdot)$ is a proper, lower semicontinuous, $\mu$–strongly convex function for every $x \in X$;*

    *(A2)  $\Psi(\cdot, y)$ is continuously differentiable on $X$ for every $y \in Y$;*

*Moreover, define the functions*

$$\psi(x) := \max_{\widetilde{y} \in Y} \Psi(x, \widetilde{y}), \quad y(x) := \operatorname*{argmax}_{\widetilde{y} \in Y} \Psi(x, \widetilde{y}), \tag{18}$$

*for every $x \in X$. If $Y$ is bounded then:*

    *(a)  $y$ is continuous on $X$;*

    *(b)  $\psi$ is continuously differentiable on $X$ and $\nabla \psi(x) = \nabla_x \Psi(x, y(x))$ for every $x \in X$.*

We now show Proposition 1 in the main paper, restated below for convenience.

**Proposition.** *(Proposition 1 in the main paper) Suppose $g$ is 1–strongly convex and symmetric. Moreover, suppose that $\Delta_k^{n-1} \subseteq \operatorname{dom} g$. Then the following properties hold for any $z \in \mathbb{R}^n$ and $\alpha > 0$:*

    *(a)  $\lim_{t \to \infty} p_t(z) = p_\infty(z)$;*

    *(b)  for any $1 \le i, j \le n$, we have $p_\alpha(z)_i \ge p_\alpha(x)_j$ if and only if $z_i \ge z_j$;*

    *(c)  the function $p_\alpha(z)$ is $\alpha$–Lipschitz continuous.*

*Proof.* For ease of notation, define

$$\Psi(x, y) := \langle x, y \rangle - \frac{1}{\alpha} g(y) \quad \forall (x, y) \in \mathbb{R}^n \times \mathbb{R}^n,$$

denote $\psi$ and $y$ as in (18), and remark that $y = p_\alpha$.

(a) Let $\{t_n\}$ be a positive sequence of scalars tending to infinity and let $z \in \mathbb{R}^n$ be fixed. Moreover, denote $y_n = p_{t_n}(z)$ for every $n \ge 1$. Fixing $n \ge 1$, the optimality condition of $y_n$ is

$$z \in \frac{1}{\alpha_n} \partial g(y_n) + N_{\Delta_k^{n-1}}(y_n) \iff \langle z, \widetilde{z} - y_n \rangle \le \frac{1}{\alpha_n} \langle u_n, \widetilde{z} - y_n \rangle \quad \forall (u_n, \widetilde{z}) \in \partial g(y_n) \times \Delta_k^{n-1}$$

Applying the 1–strong convexity of $g$ to the latter form yields

$$\langle z, \widetilde{z} - y_n \rangle \le \frac{1}{\alpha_n} \left[ g(\widetilde{z}) - g(y_n) - \frac{1}{2} \|\widetilde{z} - y_n\|^2 \right],$$

$$\le \frac{1}{\alpha_n} \underbrace{\sup_{z, \widetilde{z} \in \Delta_k^{n-1}} \left\{ g(\widetilde{z}) - g(z) - \frac{1}{2} \|\widetilde{z} - z\|^2 \right\}}_{=:C}$$

for every $\widetilde{z} \in \Delta_k^{n-1}$. Moreover, using the finiteness of $g$ on $\Delta_k^{n-1}$ and the boundedness of $\Delta_k^{n-1}$ it is clear that the quantity $C$ above is finite. Hence, since $\lim_{n \to \infty}(C/\alpha_n) = 0$, we conclude that $y_n$ converges to a solution in $p_\infty(z)$. The conclusion now follows from the definitions of $p_\alpha$ and $p_\infty$.

(b) Let $y = p_\alpha(z)$ and $i \ne j$ be fixed. The optimality condition of $y$ is $\alpha z \in \partial g(y) + \alpha N_{\Delta_k^{n-1}}(y)$, or equivalently, there exists $\tilde{n} \in N_{\Delta_k^{n-1}}(y)$ such that

$$g(\tilde{y}) - g(y) \ge \langle \alpha(z - \tilde{n}), \tilde{y} - y \rangle \quad \forall \tilde{y} \in \mathbb{R}^n.$$

In particular, let $\tilde{y}$ be the vector $y$ but where positions $i$ and $j$ are swapped. Then, the symmetry of $g$ implies that $g(\tilde{y}) = g(y)$ and we have

$$
\begin{aligned}
0 \geq \langle \alpha z - \tilde{n}, \tilde{y} - y \rangle &= \alpha(z_i - \tilde{n}_i)(y_j - y_i) + \alpha(z_j - \tilde{n}_j)(y_i - y_j) \\
&= -\alpha(z_i - z_j)(y_i - y_j) - \alpha\tilde{n}_i(y_i - y_j) - \alpha\tilde{n}_j(y_j - y_i)
\end{aligned} \tag{19}
$$

Now, the definition of $N_{\Delta_k^{n-1}}(y)$ and the fact that $\tilde{y} \in \Delta_k^{n-1}$ imply that

$$
0 \leq -\langle \tilde{n}, \tilde{y} - y \rangle = -\tilde{n}_i(y_j - y_i) - \tilde{n}_j(y_i - y_j). \tag{20}
$$

Combining (19) and (20) yields

$$
(z_i - z_j)(y_i - y_j) \geq 0
$$

which clearly implies the desired monotonicity property.

(c) Observe first that $\psi = (\delta_{\Delta_k^{n-1}} + g/\alpha)^*$ and that Proposition 1(b) implies that $\nabla\psi = y(\cdot) = p_\alpha(\cdot)$. Since the convex conjugate of a $1/\alpha$ convex function is differentiable with $\alpha$–Lipschitz continuous gradient, and $g$ is 1–strongly convex, we conclude that $p_\alpha(\cdot)$ is $\alpha$–Lipschitz continuous, as required. $\qquad\square$

## C  Projections onto the truncated simplex

We prove a key result from the main paper.

**Theorem** (Theorem 2 in the main paper). *Suppose $g(z) = \sum_{i=1}^n h(z_i)$ for every $z \in \mathbb{R}^n$, where $h : \mathbb{R} \mapsto \mathbb{R}$ is 1–strongly convex, differentiable, and its derivative $h'$ is surjective. Then, there exists $\mu \in \mathbb{R}$ such that*

$$
x = p_\alpha(z) \quad \text{if and only if} \quad
\begin{cases}
\forall i, & x_i = \Pi_{[0,1]}(y_i(\mu)), \\
\sum_{i=1}^n x_i = k,
\end{cases} \tag{21}
$$

*where $\Pi_{[0,1]}$ is the projection onto the interval $[0,1]$ and*

$$
y_j(\mu) = (h')^{-1}(\alpha z_j - \mu).
$$

*Proof.* Let $z \in \mathbb{R}^n$ be fixed. Moreover denote $\phi = (h')^{-1}$ and $p = p_\alpha(z)$. Since $p$ can be expressed in terms of a convex programming problem with nonempty relative interior, the KKT conditions are both necessary and sufficient to characterize it. These conditions, in particular, can be written as: that there exist multipliers $\lambda^\ell, \lambda^u \in \mathbb{R}_+^m$ and $\mu \in \mathbb{R}$ such that

$$
p_j = \phi(\alpha(z_j + \lambda_j^\ell - \lambda_j^u) - \mu), \quad \sum_{i=1}^n p_i = k, \quad 0 \leq p_j \leq 1, \tag{22}
$$

$$
\lambda_j^\ell p_j = 0, \quad \lambda_j^u(1 - p_j) = 0, \tag{23}
$$

for every $j \in \{1, ..., n\}$. Let now fix $j \geq 1$. Let $p_j$ satisfy (22) and (23). We wish to show that $p_j = \Pi_{[0,1]}(\phi(\alpha z_j - \mu))$. First, if $p_j = 0$ then (23) implies that $\lambda_j^u = 0$, and the fact that $\phi$ is monotonically increasing, (22), and the nonnegativity of $\lambda_j^\ell$ imply that

$$
\phi(\alpha z_j - \mu) \leq \phi(\alpha(z_j + \lambda_j^\ell - \lambda_j^u) - \mu) = p_j = 0.
$$

Second, if $p_j = 1$ then (23) implies that $\lambda_j^\ell = 0$, and the fact that $\phi$ is monotonically increasing, (22), and the nonnegativity of $\lambda_j^u$ imply that

$$
\phi(\alpha z_j - \mu) \geq \phi(\alpha(z_j + \lambda_j^\ell - \lambda_j^u) - \mu) = p_j = 1.
$$

Third, if $p_j \in (0,1)$ then $\lambda_j^\ell = \lambda_j^u = 0$ and $p_j = \phi(\alpha z_j - \mu)$. Combining all three cases, and the using the definition of $\phi$ and $p$, yields the the characterization in (21).

Conversely, suppose that $p_j = \Pi_{[0,1]}(\phi(\alpha z_j - \mu))$ satisfying $\sum_{i=1}^n p_j = k$ holds. We wish to obtain $\lambda_j^u, \lambda_j^\ell$ such that (22) and (23) are satisfied. Let $q_j = \phi(\alpha z_j - \mu)$. First, if $q_j \in (0,1)$ then clearly $\lambda_j^u = 0$ and $\lambda_j^\ell = 0$ suffice. Second, if $q_j \leq 0$, then the monotonicity of $\phi$ implies that $\phi^{-1}(0) \geq \alpha z_j - \mu$, and hence, $\lambda_j^u = 0$ and $\lambda_j^\ell := [\phi^{-1}(0) + \mu]/\alpha - z_j \geq 0$ suffice. Third, if $q_j \geq 1$, then the monotonicity of $\phi$ implies that $\phi^{-1}(1) \leq \alpha z_j - \mu$, and hence, $\lambda_j^\ell = 0$ and $\lambda_j^u := -[\phi^{-1}(1) + \mu]/\alpha + z_j \geq 0$ suffice. $\qquad\square$

# D  Proof of Proposition 3

Applying Theorem 2 to the Euclidean projection case where $h$ is the identity, we have that the solution of the projection is given by $p_\alpha(z) = \Pi_{[0,1]}(\alpha(z_i - \mu))$, where $\mu$ satisfies the condition

$$\pi(\alpha, \mu) := \Pi_{[0,1]}(\alpha(z_i - \mu)) = k.$$

Thus, to prove the first part of the proposition, it suffices to prove that there exists $\alpha$ such that $\pi(\alpha, \mu_y) = k$. Recall that $\mu_y = \min(z_y, z_{[k]}) - \eta$ and thus the vector $z - \mu$ has at least $k$ positive entries. It follows that $\pi(\alpha, \mu_y) \geq k$ for $\alpha$ large enough. On the other hand, we have $\pi(0, \mu_y) = 0$ and that $\alpha \mapsto \pi(\alpha, \mu_y)$ is continuous. Hence, by the Intermediate Value theorem, there exists $\alpha_y > 0$ such that $\pi(\alpha_y, \mu_y) = k$, as desired.

To prove the second part of the theorem, let $s = |\{i : \alpha_y(z_i - \mu_y) > 0\}|$ as well as $t = |\{i : \alpha_y(z_i - \mu_y) > 1\}|$. Then, we have

$$p_\alpha(z)_{[i]} = \begin{cases} 1 & i \in [1, t] \\ \alpha_y(z_{[i]} - \mu_y) & i \in [t+1, s] \\ 0 & i \in [s+1, n] \end{cases}$$

and the condition $\pi(\mu_y, \alpha_y) = k$ becomes $t + \sum_{i=t+1}^{s} \alpha_y(z_{[i]} - \mu_y) = k$, i.e.,

$$\alpha_y = \frac{k - t}{\sum_{i=t+1}^{s}(z_{[i]} - \mu_y)}. \tag{24}$$

Finally, to compute $t$, observe that by definition, $t$ is the unique index in $[0, \min(s, k))$ such that $\alpha_y(z_{[t]} - \mu_y) > 1$ and $\alpha_y(z_{[t+1]} - \mu_y) \leq 1$, where, by convention, $z_{[0]} = +\infty$. That is, $t$ satisfies the condition

$$\frac{\sum_{i=t+1}^{s}(z_{[i]} - \mu_y)}{k - t} \in \left[z_{[t+1]} - \mu_y, z_{[t]} - \mu_y\right). \tag{25}$$

Thus it suffices to try all values of $t \in [0, \min(s, k))$, and check condition (25). Checking the condition for all $t$ can be done as follows:

---
**Algorithm 1** Computing the index $t$ in Proposition 3
---
1: **Input:** $z \in \mathbb{R}^n, y \in \{1, \ldots, n\}$.
2: $\mu_y \leftarrow \min(z_{[k]}, z_y) - \eta$
3: Find the $k$ largest components $z_{[1]}, \ldots, z_{[k]}$.
4: $t \leftarrow 0$.
5: $S \leftarrow \sum_{i=1}^{n} \Pi_{[0,1]}(z_i - \mu_y)$.
6: **while** $\frac{S}{k-t} \notin [z_{[t+1]} - \mu_y, z_{[t]} - \mu_y)$ **do**
7:     $t \leftarrow t + 1$.
8:     $S \leftarrow S - z_{[t]}$.
9: **Return** t

---

We now make a few remarks. First, since $t$ can be at most $k$, we only need to compute the $k$ largest components (step 3), which costs $\mathcal{O}(n \log k)$. Second, the initial value of the sum $S$ (step 5) is equal to $\sum_{i=1}^{s} z_{[i]} - \mu_y$, but summing the projections $\Pi_{[0,1]}(z_i - \mu_y)$ allows us to avoid sorting the vector $z$. Third, in view of the first and second remarks, the total complexity of the algorithm is $\mathcal{O}(n \log k)$.

# E  Numerical Experiments: Detailed Description

In the Movielens datasets, each training example is a (user, rated movie) pair.[2] In our notation, $\mathcal{X}$ represents the set of users, $\mathcal{Y} = \{1, \ldots, n\}$ represents the set of movies, and $Y_x \subset \mathcal{Y}$ represents the set of movies rated by user $x$.

The output of the model is given by $z_\theta(x)_y = \langle \theta_x, \theta_y \rangle$, the dot product of a user embedding vector $\theta_x \in \mathbb{R}^d$ and a movie embedding vector $\theta_y \in \mathbb{R}^d$, which is similar to a matrix factorization model. The model parameters are $\theta = \{\theta_x : x \in \mathcal{X}\} \cup \{\theta_y : y \in \mathcal{Y}\}$. The dimensions of these vectors are fixed for all experiments to $d = 35$ for ML100K and ML20M, and to $d = 100$ for ML1B.

Our approach can be used in more complex models where the score $z_\theta(x)_y$ can be a function of other features beyond the user index $x$ and the movie index $y$. We decided to focus on a simple model class, since it is the most commonly studied for the Movielens datasets, and comparative studies show that it achieves state of the art results [6].

In each of the three datasets ML100K, ML20K, and ML1B, the set of (user, rated movie) pairs has been randomly partitioned into train (80%), cross-validation (10%), and test (10%). The training set is used to learn the parameters $\theta$ of each model, the cross-validation set is used to tune the hyperparameters of the algorithm, and the final assessment of the model quality is computed on the test set. The hyper-parameters considered are the following:

- A parameter $\sigma$ controlling the standard deviation of the initial parameter distribution. Each embedding $\theta_x, \theta_y \in \mathbb{R}^d$ is initialized from a truncated Gaussian of mean 0 and standard deviation $\frac{\sigma}{d}$, so the squared Euclidean norm of each initial vector is equal to $\sigma$ in expectation.
- The learning rate $\nu$. All models are trained using stochastic gradient descent with a constant learning rate.
- A regularization factor $\rho$ that multiplies the $L_2$ regularization term $\|\theta\|_2^2$.

All experiments use a training batch $B$ of size $|B| = 10{,}000$, $1000$, and $10{,}000$ for ML100K, ML20M, and ML1B, respectively. Each example in the batch consists of a (user, rated movie) pair $(x, y)$. The loss function on the batch is given by

$$\frac{1}{|B|} \sum_{(x,y) \in B} \ell(z_\theta(x), y) + \rho(\|\theta_x\|_2^2 + \|\theta_y\|_2^2) \, , \tag{26}$$

where $\ell$ is either $\ell_{\text{Softmax}}$, $\ell_{\text{Rankmax}}$ defined in Section 3.3, or $L_{\text{sparsemax}}$ as defined in Equation (20) in [5]. Note that each loss $\ell$ involves a sum over all labels $y \in \mathcal{Y}$. In particular, the sum appears in the denominator of Rankmax and Softmax, as well as the quadratic term in the definition of $L_{\text{Sparsemax}}$. We compute the full sum on ML100K and ML20M datasets, and approximate this sum for ML1B dataset due to its large scale. More specifically, for each batch in our sampling approach, we use a random sample of 10,000 movies following a frequency-based distribution [1] in which each movie $y$ is sampled with probability $\propto f_y^{-0.5}$, where $f_y$ is the frequency of movie $y$ in the training data.

As common in the evaluation of retrieval models, when we compute metrics on the test set, the positive labels seen during training are excluded from the computation, so that they are treated neither as positives nor as negatives. For example, in the computation of Precision@K in equation (7), if $Y_x$ is the set of all positive labels for an example $x$, and $Y_x$ is partitioned into $Y_x^{\text{train}}, Y_x^{\text{validation}}, Y_x^{\text{test}}$, then the metric on the test set is computed using a ranking over $\mathcal{Y} \setminus (Y_x^{\text{train}} \cup Y_x^{\text{validation}})$ and the set of positive labels to be $Y_x^{\text{test}}$.

The results are given as a function of number of epochs in Figure 2 in the main paper, and as a function of wall-time in Figure 3 below. Our implementation of Sparsemax is slower than Rankmax and Softmax, mainly because it requires sorting the vector of scores $z_\theta(x)$.

In our experiments with ML100K, models are trained on a large number of epochs (20,000) to the point where they over-fit the training data. The set of best hyperparameters is determined by the maximum Precision@10 reached on the cross-validation set, and the training iteration at which this maximum value is reached is recorded for early stopping. The values reported on Table 2 are the ranking metrics measured on the test set, averaged over a window centered around the early stopping number of training iterations. The best hyper-parameters found for ML100K are $\sigma = 0.1, \nu = 1, \rho = 0.06$ at epoch 9400, $\sigma = 0.1, \nu = 1, \rho = 0.1$ at epoch 7500, and $\sigma = 0.1, \nu = 4, \rho = 0.04$ at epoch 9500, for Rankmax, Softmax, and Sparsemax, respectively. The plots for ML100K in Figure 2 and 3 include smoothing under a rolling window of size 10.

In contrast, for ML20M we carried out hyperparameter search while training models for 30 epochs and did not observe much over-fitting. All values in Table 2 are therefore metrics measured on test after 30 epochs. The best hyper-parameters found for ML20M are $\sigma = 0.1, \nu = 4, \rho = 0.0125$,

$\sigma = 0.1, \nu = 4, \rho = 0.0125$, and $\sigma = 0.1, \nu = 12, \rho = 0.0125$, for Rankmax, Softmax, and Sparsemax, respectively. Figure 2 and 3 shows these metrics (no smoothing applied) on test after 50 epochs to illustrate the trend after the 30 epoch mark.

On the larger ML1B dataset, models are trained for 20 epochs and some over-fitting is observed. Since the metric R@1000 shows some noise, the metric includes smoothing under a rolling window and the maximum values on the test set are reported in Table 2, with $\sigma = 0.1, \nu = 4, \rho = 0.012$ for both Rankmax and Softmax losses.

Figure 3: Retrieval metrics vs. wall time on MovieLens 100K and 20M.

## Footnotes

[2]The original training data also includes a rating value, but the model's task consists in predicting whether a movie is rated, and not the numerical value of the rating. This is often called the implicit feedback setting [3], commonly used in retrieval models.