[Reviews · NeurIPS 2020]

Review 1

Summary and Contributions: This paper proposes rankmax which can be seen as an adaptive version of softmax. The adaptive refers to the fact that each training example is treated differently during training. Results show that rankmax outperforms non-adaptive projection methods significantly.

Strengths: - I think that adaptive parametrisation to individual training examples is interesting. - Theoretical justification is solid.

Weaknesses: 1. Overall I find that it is hard to follow the paper. I think writing and presentation should be improved. I read several math-heavy machine learning books. But still this paper is difficult to follow. This paper is written in way that only particular people understand. 2. Lack of experiments. Since authors claim the proposed method is a good candidate for retrieval problems, they should consider conducting experiments on a vision task such as Stanford Online Products (SOP) [a]. [a] Cross-Batch Memory for Embedding Learning, CVPR 3. No visualisation is provided. Where rankmax does better job than softmax during testing?

Correctness: It seems they are correct.

Clarity: I can understand easily the essence method and claims. However, if I want to understand more deeply, only people who work on this field understand. I am sorry that I cannot give a suggestion how to improve the writing. Thanks.

Relation to Prior Work: I like this part. Connection to prior work is clear.

Reproducibility: No

Additional Feedback: After reading rebuttal and the other reviewers, I would like to increase +1. Thanks for the rebuttal.


Review 2

Summary and Contributions: The paper focuses on the design of Lipschitz continuous approximation of the top-k argmax function, with parameter adaptable to each training sample. It proposes a general setting of projecting a score vector on the (n,k)-simplex which may prove useful for retrieval tasks. Thereon the Rankmax function is derived with adaptation scheme which depends on the label of the training sample. The approach is further illustrated on a recommendation system application and compared to Softmax and Sparsemax to show its effectiveness.

Strengths: * The paper is concerned with the derivation of k-argmax function's continuous approximation as a generic projection of a score vector onto the (n, 1)-simplex or the (n, k)-simplex (for predicting top-k relevant labels) based on a strongly convex function $g$. The first interesting contribution shows how to obtain such approximation and derives the general solution of this problem provided some properties of $g$ (it is separable, 1-strongly convex). Relevant $g$s are quadratic function, negative entropy. * The second contribution considers using the projected score vector on the (n,k)-simplex in a cross entropy loss for retrieval or multi-label classification tasks as a proxy of Precision@k metrics. Specifically, Euclidean projection with adapted Lipschitz constant $\alpha$ of the projection to the training instance is devised as the Rankmax operator. The key element is that $\alpha$ can be computed such that the sample's labels occurs in the top-k. Also the projection is sparse contrary to the softmax operator, so does the gradient of the loss function. * Empirical evaluations on recommendation applications are proposed to investigate how the Rankmax function performs compared to Softmax and Sparsemax. The experiments were carried out over medium to large scale datasets. The experimental setup is well described and sound. The reported performances shonw that Rankmax is on par with the other functions on medium datasets and significantly outperform them for large scale ones. * Overall, the proposed results related to the generic projection on the (n,k-simplex are significant and relevant. The theoretical motivations and derivations are well grounded and well exposed in the paper.

Weaknesses: * The paper advocates in the conclusion the link of the Rankmax function to the OWA (ordered weighted Average) operator [1] which focuses on the top ranked elements in ranking problems. Although, OWA's framework does not relate to projection on (n, k)-simplex, the authors should discuss their connections as the end claim of the paper is an adaptive projection for learning top-k classification model. As a general remark, the paper cannot not avoid discussing related work on top-k prediction methods. * The overall optimization algorithm based on the cross-entropy objective function (13) deserves to be discussed. Also I wonder if Rankmax will not result in overfitting (especially in online learning setting) as the adaptation scheme according to the positive label ensures that the positive label is always well predicted and this may be detrimental if the training set is noisy. * While comparing the max operators in Figure 2, (mostly bottom panel), it will be more relevant to show the performances along the computation time (instead of the epochs) as all tested max approximations tend to produce similar performances as the number of epochs goes on. This will enable to assess the benefit of using Rankmax in terms of balance between accuracy and computation burden. Such as, the Rankmax function appears to have little impact on the performances on the large scale dataset. * It is stated the the metrics for smaller $K$ are noisy. How do they look like? Also numerical experiments on multi-class classification (the simplest case of Rankmax) would certainly improve the paper.

Correctness: The main results presented in the paper are correct (except some remarks reported in section 6 of the review). The empirical methodology is clear and follows the standards of machine learning.

Clarity: * Theorem 2: it is supposed that $g$ is $\mu$-strong convex. This might be confusing with the dual variable $\mu$ characterizing the solutions. * Line 157: the function $r(z)_y$ is not defined $ Line 158: the statement $p_\alpha(z)_y =cc $ for all $y \in Y_x$ is unclear and should be better explained * Equation (15): there is a $s$ appearing in the sub-differential of the log Rankmax function which is not defined. $\ell_\text{softmax}(z)$ function should be appended the label $y$ as for the Rankmax counterpart. The derivative should write $-Softmax(z)_i$ for $i \neq y$ according to the equation in line 217. * Line 216: the expression of log Rankmax seems to be incorrect. * In Equation 14 as there is only one label of interest, we do no required the permutation of $z$ (that is the terms $z_[i]$ at the denominator as all $z_i$ are shifted by the same amount $z_y - 1$. * Equation 13: \alpha_y and \mu_y are quantities that depends on the model parameter $\theta as they are computed from $z_\theta(x)$. This should be made clear in the formulation.

Relation to Prior Work: The paper discussed relation to prior work related to projection on the (n, k)-simplex. However the final application of the derived Rankmax being the learning of classification model able to predict appropriately top-k labels, authors should relate their work to top-k ranking methods. The brief discussion on the relation to [1] should be expanded. Also I shall mention a recent paper on differentiable ranking using optimal transport [2]. References [1] Usunier, N., Buffoni, D., & Gallinari, P. (2009, June). Ranking with ordered weighted pairwise classification. In Proceedings of the 26th annual international conference on machine learning (pp. 1057-1064). [2] Cuturi, M., Teboul, O., & Vert, J. P. (2019). Differentiable ranking and sorting using optimal transport. In Advances in Neural Information Processing Systems (pp. 6861-6871).

Reproducibility: Yes

Additional Feedback: After rebuttal -------------- The feedback addresses most of the concerns raised in the review. - Connections of Rankmax to top-k classification will be thoroughly discussed - When used in online setting, the rankmax does not involve an overfitting according to the authors' feedback. - An effort has to be put to discuss the overall optimization algorithm based on the cross-entropy objective function (13)


Review 3

Summary and Contributions: [EDIT: thank you for the thoughtful consideration of my comments and questions. I am maintaining my positive assessment.] This paper presents a sparse projection onto the so-called (n,k)-simplex, aka. the capped simplex {p: 0 <= p_i <= 1 and <1,p>=k}, used in an adaptive fashion to obtain a likelihood loss. The paper develops some useful results along the way and results in well-performing models evaluated on recsys tasks.

Strengths: - Useful, generic results for penalized projections onto this set, which could be useful for future work. - Creative motivation of an adaptive "temperature" parameter to guarantee that all positive labels have nonzero mass. - Efficient resulting algorithm if n >> k.

Weaknesses: - The paper misses some important closely-related results and applications. - Empirical evaluation limited to a recsys setting (while baselines are not particularly specialized for recsys), more standard multi-label / ranking baselines would be interesting too (see tasks in related work listed below). - Not justified why we should jump through these hoops to support a likelihood loss over alternatives (e.g. a Fenchel-Young loss characterized by the polytope -- see references below.)

Correctness: The results developed in the paper seem correct and the empirical methodology is correct as well.

Clarity: The paper is well written and informative.

Relation to Prior Work: The paper omits relevant related work. In [1,2] projections onto the exact same polytope are studied, and [1] even uses a dense projection with a likelihood loss and several applications that should be of interest. In [3] a general framework for constructing loss functions on such polytopes is discussed and analyzed, such a top-k Fenchel-Young loss would be an interesting point of comparison here to contrast against the likelihood loss. Proposition 5 in [3] also gives a general bisection-based algorithm that is very similar to your Theorem 2. Their extended version [4] discusses the connection between the capped simplex and the permutahedron, suggesting the applicability of the algorithms from [5]. Finally, [6] studies multi-label classification with the same polytope from a structured prediction perspective. [1] The Limited Multi-Label Projection Layer. B. Amos, V. Koltun, and J. Z. Kolter, arXiv:1906.08707, 2019 [2] Weiran Wang and Canyi Lu, Projection onto the Capped Simplex, arXiv preprint arXiv:1503.01002, 2015 [3] Mathieu Blondel, André F. T. Martins, Vlad Niculae. Learning Classifiers with Fenchel-Young Losses: Generalized Entropies, Margins, and Algorithms, In: AISTATS 2019 [4] Mathieu Blondel, André F.T. Martins, Vlad Niculae; 21(35):1−69, 2020. Learning with Fenchel-Young losses. JMLR 21(35):1−69, 2020. [5] Cong Han Lim, Stephen J. Wright. Efficient Bregman Projections onto the Permutahedron and Related Polytopes. In: AISTATS 2016. [6] Mathieu Blondel. Structured Prediction with Projection Oracles. In NeurIPS 2019.

Reproducibility: Yes

Additional Feedback: Despite the missing references, I think the mapping proposed in this paper can be of interest. The connection to pairwise losses and the WARP loss briefly discussed in the conclusions setting is very interesting, consider expanding upon that direction! It would be interesting to explore the relationship between the adaptive alpha guaranteeing probability mass on the true labels and the separation margin property as hinted by your phrasing. Note that the sparsemax loss has a separation margin, similar to the hinge loss, despite not necessarily giving mass to all true labels. There is more work on the separation margins of FY losses in [3]. While the n log n sorting-based algorithm for sparsemax may be impractically slow for the large dataset you consider, you could use a root finding algorithm (your Theorem 2, or Prop. 5 in [3]) to get numerically accurate gradients with T*n cost, where T=18 iterations should be sufficient for float32 machine precision.


Review 4

Summary and Contributions: This paper studies an adaptive projection alternative to the softmax function that is based on a projection on the (n,k)-simplex. The authors explicit the solution of this projection prior to study the use of it in the context of multi-class classification. They further introduce an adaptive version of the projection, optimized for each example.

Strengths: The projection framework is neat, sound and well developed.

Weaknesses: One of the major drawback of this study is the lack for strong justification of the projection on the (n,k)-simplex (beyond references to previous studies). Indeed, the justification given for its use in multi-class classification is not entirely convincing as it seems possible to have a classifier that has maximal precision (and thus is optimal acc. to (8)) while not being optimal acc. to (9). I believe the paper would be more convincing if the authors provide this justification and apply their method on top of other approximations (typically differentiable approximations of (8)).

Correctness: As far as I can tell, the claims are correct. The empirical methodology is overall sound. Comparisons with

Clarity: The paper is well written and agreeable to read.

Relation to Prior Work: The relation to prior art is well described and discussed as far as I can tell.

Reproducibility: Yes

Additional Feedback:

[Author Response · NeurIPS 2020]

## Foreword

We would like to thank the reviewers for their valuable feedback and suggested references. Two of the reviewers raised the question of whether the use of the cross-entropy loss is justified, and we would like to address this point. First, Rankmax can be thought of as an adaptive projection method that is not tied to a particular loss function, and it can be used with other losses such as Fenchel-Young losses or differentiable approximations of top-$k$ losses – we thank the reviewers for these references. On the other hand, we focused on the cross-entropy loss because it is commonly used with the closely related Softmax projection. Moreover, we have shown that in the cross-entropy setting, there are useful connections between Rankmax and other rank losses such as OWA and pairwise losses (see Section 3.3 and Section 5). Note that our purpose is not to make a statement about which loss is better suited to multi-label classification, but rather to show that adaptivity of the Rankmax projection can accelerate training and improve model quality.

## Detailed answers

**(R1–R4)** We will correct all typos, missing citations, spelling mistakes, and minor technical errors in the final revision.

**(R1)** *Mathematical notation and writing.* Section 2 and 3 follow the notational conventions of the field of optimization. Nevertheless, we tried to make the core findings and claims of the paper accessible to a wider audience.

**(R2)** *Connections to top-$k$ classification.* Thank you for the suggestions. We will expand our discussion of the connection between Rankmax and top-$k$ classification, both in Section 3 and when discussing the gradient properties. In particular, it can be shown that the magnitude of the gradient under OWA only depends on the number of negatives above the margin, while it is adaptive under Rankmax (it depends on the distribution of negative scores). This gives another interpretation of adaptivity. We will give specific examples in which the magnitudes can vary significantly.

**(R2)** *Problems with overfitting.* Based on our experimental results, Rankmax was prone to overfitting on the smaller dataset (as discussed line 269), but this can be remedied using early stopping on a cross-validation set. With early stopping, it outperformed non-adaptive projections. On the larger datasets, we have not observed any overfitting.

**(R2)** *Compute time vs. epochs.* We will add wall-time plots to the supplement. As briefly discussed Line 257, Rankmax and Softmax had the same computational cost per epoch, but Sparsemax was slower.

**(R3)** *Projection on the $(n,k)$-simplex (or capped simplex).* Thank you for these additional references, we will correct this omission in the revision. For comparison, [1,2] consider special cases of projections on the capped simplex (respectively for Euclidian and entropy regularizers) and [3] considers the projection on the standard simplex. Our result (Theorem 2) can be viewed as a generalization of both. The permutahedron projection in [4] can indeed be applied to the capped simplex, though the result in Theorem 2 is much more direct to obtain, and easier to interpret and to implement (as it is more specialized). We will add a detailed discussion in the related work section.

**(R3)** *Fenchel-Young (FY) losses.* Thank you for bringing this important work to our attention. We will add a discussion of FY losses in Section 3. As discussed above, though we focused the presentation on cross-entropy, the Rankmax projection can be used with other losses. Combining the adaptive projection of Rankmax with FY losses is an interesting direction for future work. Specifically, the FY loss with regularizer $\alpha g$, label $y$ and score vector $z$, can be written as $L(z,y) = f_z(y) - f_z(p_\alpha(z))$ where $p_\alpha(z) = \arg\min_p -\langle z, p \rangle + \alpha g(p)$ is the projection defined in Eq. (1) of our paper. It is therefore possible to apply the same loss with adaptive $\alpha$.

**(R4)** *Relevance of projecting onto the (n,k)-simplex.* The adaptivity of Rankmax projection is the central contribution in the paper. We developed our framework in the more general context of projecting onto the $(n, k)$-simplex because it was found useful in other studies (see [1,5,6]). However, Rankmax provides benefits even when $k = 1$, as shown in our numerical experiments.

**(R4)** *Differentiable approximations of Precision@k.* Thank you for the suggestion, we will add in Section 3 a discussion of differentiable approximations to top-$k$ metrics. As discussed above, Rankmax is an adaptive projection that can be used with any loss function, including such approximations. We focused on the cross-entropy loss as it enjoys additional properties and connections with other losses.

[1] Amos, Koltun, and Kolter. The Limited Multi-Label Projection Layer, arXiv:1906.08707, 2019.
[2] Wang and Lu. Projection onto the Capped Simplex, arXiv:1503.01002, 2015.
[3] Blondel, Martins, and Niculae. Learning Classifiers with Fenchel-Young Losses: Generalized Entropies, Margins, and Algorithms. AISTATS, 2019.
[4] Lim and Wright. Efficient Bregman Projections onto the Permutahedron and Related Polytopes. AISTATS, 2016.
[5] Lapin, Hein, and Schiele. Top-k multiclass SVM. Advances in Neural Information Processing Systems, 2015.
[6] Lapin, Hein, and Schiele. Analysis and optimization of loss functions for multiclass, top-k, and multilabel classification. IEEE transactions on pattern analysis and machine intelligence, 2017.


[Meta-Review · NeurIPS 2020]

Three knowledgeable referees support acceptance for the contributions. One reviewer (R1) was slightly on the reject side but I discounted that review because of low confidence. Therefore I recommend acceptance. However, please consider revising your paper to include suggested references, as also promised in the rebuttal, and if possible also extend your empirical evaluation.